palaeontology

Diapsida, feeding apparatus, *Colobops noviportensis*, Rhynchocephalia, Triassic

**Author for correspondence:**
Torsten M. Scheyer
e-mail: tscheyer@pim.uzh.ch

# *Colobops*: a juvenile rhynchocephalian reptile (Lepidosauromorpha), not a diminutive archosauromorph with an unusually strong bite

Torsten M. Scheyer[1], Stephan N. F. Spiekman[1],
Hans-Dieter Sues[2], Martín D. Ezcurra[3,4],
Richard J. Butler[4] and Marc E. H. Jones[5]

[1]Universität Zürich, Paläontologisches Institut und Museum, Karl Schmid-Strasse 4, Zurich CH-8006, Switzerland
[2]Department of Paleobiology, National Museum of Natural History, Smithsonian Institution, MRC 121, Washington, DC 20560, USA
[3]Sección Paleontología de Vertebrados, CONICET-Museo Argentino de Ciencias Naturales, Ángel Gallardo 470, C1405DJR, Buenos Aires, Argentina
[4]School of Geography, Earth and Environmental Sciences, University of Birmingham, Edgbaston, Birmingham B15 2TT, UK
[5]Research Department of Cell and Developmental Biology, University College London, Gower Street, London WC1E 6BT, UK

TMS, 0000-0002-6301-8983; SNFS, 0000-0002-3197-8752;
H-DS, 0000-0002-9911-7254; MDE, 0000-0002-6000-6450;
RJB, 0000-0003-2136-7541; MEHJ, 0000-0002-0146-9623

Correctly identifying taxa at the root of major clades or the oldest clade-representatives is critical for meaningful interpretations of evolution. A small, partially crushed skull from the Late Triassic (Norian) of Connecticut, USA, originally described as an indeterminate rhynchocephalian saurian, was recently named *Colobops noviportensis* and reinterpreted as sister to all remaining Rhynchosauria, one of the earliest and globally distributed groups of herbivorous reptiles. It was also interpreted as having an exceptionally reinforced snout and powerful bite based on an especially large supratemporal fenestra. Here, after a re-analysis of the original scan data, we show that the skull was strongly dorsoventrally compressed post-mortem, with most bones out of life position. The cranial anatomy is consistent with that of other rhynchocephalian lepidosauromorphs, not rhynchosaurs. The 'reinforced snout' region and the 'exceptionally enlarged temporal region' are preservational artefacts and not exceptional among clevosaurid rhynchocephalians. *Colobops* is thus not a key taxon for understanding diapsid feeding apparatus evolution.

# 1. Background

The Triassic represents a period of major ecosystem change following the end-Permian mass extinction crisis, during which morphological innovation and evolutionary novelties occurred in many vertebrate lineages. This is especially apparent among diapsid reptiles, among which highly divergent body plans, feeding strategies as well as lifestyles evolved. These include fish-like ichthyosaurs, marine sauropterygians, chameleon-like drepanosaurs, extremely long-necked tanystropheids, herbivorous rhynchosaurs, pterosaurs, dinosaurs (which later gave rise to birds) and the distant ancestors of crown crocodylians and squamates [1–4]. Among the wealth of Triassic taxa is a small (2.5 cm long) reptile skull (YPM VPPU 18835) from the Upper Triassic New Haven Formation of Connecticut, USA. It was initially described as an indeterminate extinct member of the lepidosaurian clade Rhynchocephalia [5]. The specimen is still largely embedded in matrix, so that the skull bones are only partially exposed and their exact relations with each other were originally difficult to assess. The skull was severely damaged when it was exposed by an explosion during major road construction. The original description mentioned the crushed nature of the skull and the presence of incisor-like teeth in the tip of the rostrum (similar to those of the tuatara *Sphenodon punctatus*), which were lost during initial preparation, but which underscored at the time the original identification of the species as a member of Rhynchocephalia [5].

A recent re-study of YPM VPPU 18835 was performed using micro-computed tomography (µCT) scanning and the known specimen was formally designated as the holotype of the new genus and species *Colobops noviportensis* [6]. This analysis yielded two major results: (i) *Colobops noviportensis* is a member of Rhynchosauria (sister to all remaining rhynchosaurs) within Archosauromorpha instead of Rhynchocephalia within Lepidosauromorpha; and (ii) although small, the skull shows a reinforced snout region and greatly enlarged supratemporal fenestrae that could house powerful adductor jaw musculature [6]. Whereas an enlarged supratemporal fenestra and linked powerful bite would indeed be outstanding features among small-bodied Triassic diapsids, the new proposed cranial configuration would be truly unique, thus warranting confirmation.

The authors [6] cited five characters as diagnostic for *Colobops noviportensis*. For two of those characters, the authors indicated that the given state might reflect immaturity (character 1: 'prominent, symmetrical fontanelle between frontals and parietals in midline'; character 4: 'dorsal exposure of postorbital transversely broad, with posteriorly directed process near the transverse midpoint of the supratemporal fenestra'). Character 4, in particular, indicates that the squamosal and postorbital were interpreted as being in an anatomically correct position, indicating an enlarged supratemporal fenestra; similarly, the flattened skull profile and oblong orbits were considered genuine anatomical features rather than preservational artefacts.

Due to the rather unconventional cranial configuration figured and described by Pritchard and colleagues [6], and given that three-dimensional (3D) virtual models are not images of the raw data (in this case of the fossilized skull) but always interpretations of the dataset used [7,8], we here perform another virtual reconstruction of the original scan. This is done to see whether we can reproduce the original models and corroborate the previous findings. Based on the new results presented herein, we re-evaluate the cranial shape of *Colobops noviportensis*, its phylogenetic position, and the size and shape of its supratemporal fenestra.

# 2. Material and methods

## 2.1. Fossil and computer-tomography scan

The original µCT scan dataset of YPM VPPU 18835 used by Pritchard *et al.* [6] was obtained and segmentation of the skull bones was performed using Materialise Mimics v. 19.0. Segmentation of bone was achieved mainly manually and with a conservative approach so as to differentiate bone from matrix. In several instances, this led to omission of highly damaged or incomplete bone structures in our model, especially in the braincase and palatal regions. These omissions do not influence the general interpretation of the skull bone configuration presented herein. 3D models of the segmented bones were exported as ply files and figured in Blender 2.79. The 3D models produced here are available as supplementary packed ZIP folder.

Measurements on the 3D model were taken with Fiji [9]. In table 1, measurement 1 is used to determine the purported anteroposterior midpoint of the supratemporal fenestra. Measurements 2 (=free suprafenestral space) and 3 (=supratemporal fossa) combined equal the total width of the

**Table 1.** Measurements of the supratemporal region based on the new 3D model of YPM VPPU 18835. The measurements 1–4 are based on figure 4b, whereas the measurements in the second column indicated by an asterisk refer to an alternative reconstruction of that region as presented in figure 4c. STF, supratemporal fenestra.

| | |
|---|---|
| 1: 10.276 mm | 1*: 10.284 mm |
| 2: 6.411 mm | 2*: 6.051 mm |
| 3: 1.810 mm | 3*: 1.771 mm |
| 4: 9.663 mm | 4*: 9.110 mm |
| 4 doubled: 19.326 mm | 4* doubled: 18.220 mm |
| 2 + 3: 8.221 mm | 2* + 3*: 7.822 mm |
| ratio 'adductor chamber/total width' [(measurements 2 + 3) doubled/measurement 4 doubled]: $8.221 \times 2/19.326 = 0.851$ | ratio (measurements 2* + 3*) doubled/measurement 4* doubled: $15.644/18.220 = 0.858$ |
| STF surface area: 68.7 mm$^2$ | STF surface area: 66.8 mm$^2$ |

supratemporal fenestra. Measurement 4 approximates half of the skull width at midpoint of the supratemporal fenestra. As stated in table 1, the ratio '(measurements 2 + 3) doubled / measurement 4 doubled' provided herein equals the 'adductor chamber/total width' ratio presented by Pritchard et al. [6] in their electronic supplementary material, table S1.

Pritchard et al. [6] conducted regression analyses in which they compared the transverse width of the skull to both the transverse width of the supratemporal fossae and the proportional width of the supratemporal fossae. In both cases, they recovered *Colobops noviportensis* as an outlier with significantly enlarged supratemporal fossae. It is unclear in several instances, however, whether supratemporal fossa and supratemporal fenestra are used synonymously [6]; e.g. in their fig. 4 where 'supratemporal fossae by total skull width in modern *Iguana*' are plotted against 'log[transverse width of postorbital skull (cm)]', but the shown adult *Iguana* in their fig. 4d has a value of 1.0 at 0.75, indicating that this should be supratemporal fenestra width and not fossa width).

We used our new 3D model to revise the estimated measurements for *Colobops noviportensis*, and reran the regression analyses [6] using their measurement data (with the exception of our updated estimates for *Colobops*; note that the 'Appendix D' mentioned in their legend of electronic supplementary material, table 1 providing 'Measurement methods' is missing) and the same statistical protocols as in that study. Data for the transverse width of the skull and transverse width of the supratemporal fossae were $\log_{10}$ transformed prior to analysis. A linear regression model in R was used and confidence intervals were calculated using the predict function. The confidence intervals plotted (figure 5) match those shown by Pritchard et al. [6].

Our virtual model of the cranial structure of *Colobops noviportensis* was compared to actual specimens and 3D models of rhynchocephalians including *Sphenodon* [10,11] and *Clevosaurus* [12; see also 13]. In addition, a surface model prepared from an early juvenile specimen of *Sphenodon punctatus* from Stephens Island, New Zealand, recently published [14], was used for the study of the general skull shape, articulation of skull roof bones, and the shape of the fontanelle. The scan of this specimen (Carnegie Museum #30660) was taken under National Science Foundation grant IIS-98 to Chris Bell and is housed at DigiMorph.org, where it was made available under Creative Commons License CC BY-NC (University of Texas High-Resolution X-ray CT Facility [UTCT] Archive no. 0124). In addition, we used virtual models of the rhynchosaurs *Mesosuchus browni* (SAM-PK-6536; Middle Triassic, South Africa) and *Teyumbaita sulcognathus* (UFRGS-PV-0232-T; Late Triassic, Brazil), which were obtained from two braincase studies of those species [15,16]. The *Mesosuchus* scan of SAM-PK-6536 is housed at the Museum für Naturkunde Berlin, Leibniz-Institut für Evolutions- und Biodiversitätsforschung, Germany, and the *Teyumbaita* scan of UFRGS-PV-0232-T was obtained from C. Schultz at the Universidade Federal do Rio Grande do Sul, Brazil.

## 2.2. Phylogeny

The impact of our reinterpretations of the anatomy of *Colobops noviportensis* on its phylogenetic relationships was tested after rescoring this species in the original matrix by Pritchard et al. [6]. In order to test the phylogenetic relationships of *Colobops noviportensis*, we chose to use the most extensive phylogenetic dataset currently available for Permian and Triassic archosauromorphs [17] as modified by

subsequent authors [1,18–23]. We used this data matrix because it has key taxa and characters that we consider important to test the phylogenetic position of a putative early rhynchosaur, as is the case for *Colobops noviportensis*. For this second analysis, we modified characters 5, 39, 100, 187, 207, 351, 352, 377, 446 and 567, and added 16 characters that are phylogenetically informative, mostly among lepidosauromorphs (696–711). In addition to *Colobops noviportensis*, we added the following four lepidosauromorphs as new terminals: *Megachirella wachtleri*, *Salvator rufescens*, *Clevosaurus hudsoni* and *Sphenodon punctatus* and modified some of the scorings of the original versions of this data matrix (see electronic supplementary material, notes). This was done because the original character list of Ezcurra [17] focused on archosauromorphs and, as a result, its sampling of early lepidosauromorphs and informative characters among these taxa was limited. Thus, we added the four species-level lepidosauromorph terminals to better sample the morphological diversity in the early history of the clade. Because of the taxonomic expansion of the dataset, we also added 16 independent, informative characters for our sample of early lepidosauromorphs mostly taken from previous studies (see electronic supplementary material, information for complete overview). Ezcurra & Butler [1] scored several archosauromorphs in this data matrix with the aim of assessing morphological disparity and evolutionary rates, but not for a reconstruction of their phylogenetic relationships. Thus, we deactivated before the tree searches 35 terminals (see electronic supplementary material, notes) and character 119 (following [20]), resulting in a dataset of 121 active terminals and 710 characters.

The two matrices of discrete morphological characters were analysed under equally weighted maximum-parsimony using TNT v. 1.5 [24]. The search strategies started using a combination of the tree-search algorithms Wagner trees, tree bisection and reconnection (TBR) branch swapping, sectorial searches, Ratchet and Tree Fusing, until 100 hits of the same minimum tree length were achieved. The best trees obtained were subjected to a final round of TBR branch swapping. Zero-length branches in any of the recovered most parsimonious trees were collapsed. In the first analysis, we used the same list of additive (ordered) characters used by Pritchard *et al*. [6] during the searches. In the second analysis, the following characters were considered additive: 1, 2, 7, 10, 17, 19–21, 28, 29, 36, 40, 42, 50, 54, 66, 71, 74–76, 122, 127, 146, 153, 156, 157, 171, 176, 177, 187, 202, 221, 227, 263, 266, 278, 279, 283, 324, 327, 331, 337, 345, 351, 352, 354, 361, 365, 370, 377, 379, 386, 387, 398, 410, 424, 430, 435, 446, 448, 454, 458, 460, 463, 470, 472, 478, 482, 483, 485, 489, 490, 504, 510, 516, 529, 537, 546, 552, 556, 557, 567, 569, 571, 574, 581, 582, 588, 648, 652 and 662 because they represent nested sets of homologies. Branch support was quantified using decay indices (Bremer support values) and a bootstrap resampling analysis, using 1000 pseudo-replicates and reporting both absolute and GC ('group present/ contradicted'; i.e. difference between the frequencies of recovery in pseudo-replicates of the clade in question and the most frequently recovered contradictory clade) frequencies [25]. The minimum number of additional steps necessary to generate alternative, suboptimal tree topologies was calculated when constraining the position of *Colobops noviportensis* in different parts of the tree and rerunning the analysis.

## 2.3. Institutional abbreviations

MCN-PV, Museu de Ciências Naturais da Fundação Zoobotânica do Rio Grande do Sul (MCN/FZBRS), Porto Alegre, Brazil; MFN, Museum für Naturkunde, Leibniz-Institut für Evolutions- und Biodiversitätsforschung, Berlin, Germany; UFRGS, Universidade Federal do Rio Grande do Sul, Porto Alegre, Rio Grande do Sul, Brazil; YPM VPPU, Peabody Museum of Natural History, Yale University, Vertebrate Paleontology Princeton University Collection, New Haven, Connecticut, USA.

# 3. Results

## 3.1. Overview

The skull roof and the rostrum could be reconstructed almost completely (figure 1*a–f*), whereas most of the posterior region of the skull and the braincase elements are not preserved (figure 1*a–c*). The presence of a remnant of the left premaxilla (as indicated by Pritchard *et al*. [6], their fig. 1*a*) could not be verified (figure 1*g*). The only evidence of the premaxillae is in photographs of YPM VPPU 18835 that show the skull prior to initial preparation (figure 2). The two premaxillary teeth mentioned by Sues & Baird [5] were lost during initial preparation and the shapes of the jugals and maxillae are more complete in the original photographs. A few unidentified skull bones are also present but are not described herein.

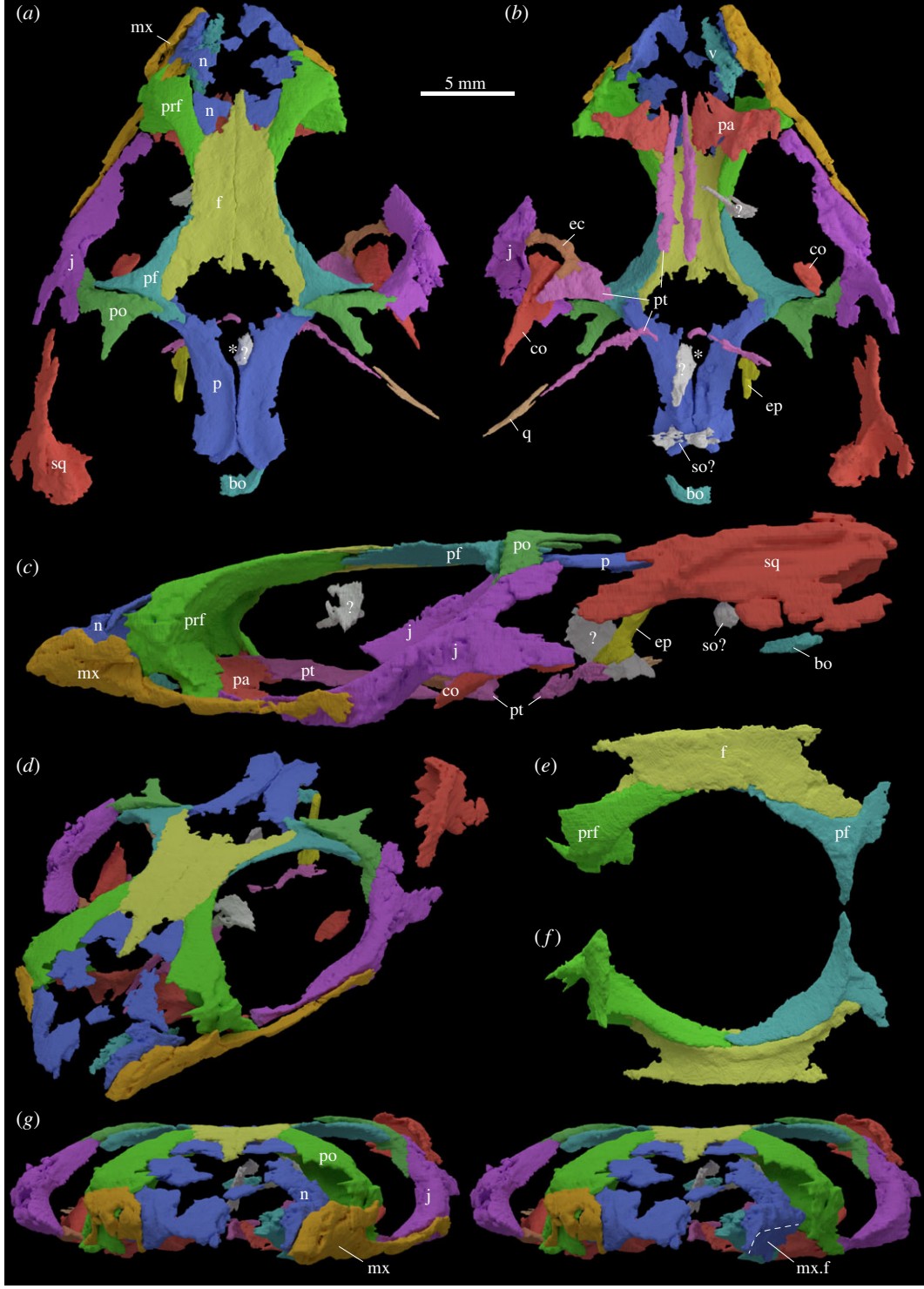

**Figure 1.** New 3D model of YPM VPPU 18835. Images in (*a*) dorsal, (*b*) ventral, (*c*) left lateral and (*d*) angled anterolaterodorsal view. Images of the articulated upper orbital rim in (*e*) dorsal and (*f*) ventral view. (*g*) Skull in anterior view with and without left maxilla, exposing maxillary facet on nasal. (*c*–*g*) Not to scale. bo, basioccipital; co, coronoid; ec, ectopterygoid; ep, epipterygoid; f, frontal; j, jugal; mx, maxilla; mx.f, maxillary facet on nasal; n, nasal; p, parietal; pa, palatine; pf, postfrontal; po, postorbital; prf, prefrontal; pt, pterygoid; so, supraoccipital; sq, squamosal; v, vomer; *, parietal foramen.

## 3.2. Skull bones

There is only one small part of the right maxilla preserved, which articulates with the nasal and prefrontal (figure 1*g*). The left maxilla is more complete with a broad but low anterior facial (=ascending) process articulating with the nasal dorsally and the prefrontal posterodorsally

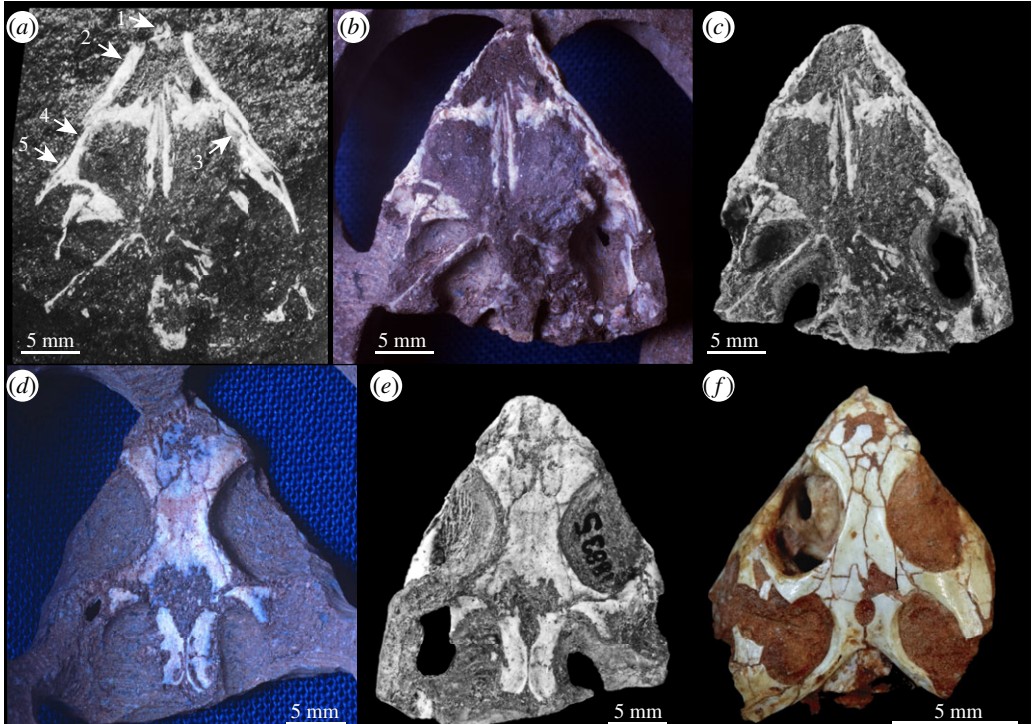

**Figure 2.** Photographs of *Colobops noviportensis* (YPM VPPU 18835) (*a–c*) ventral view; (*d,e*) dorsal view) at three different stages of preparation, in comparison to (*f*) a skull (MCN-PV 2852) of *Clevosaurus brasiliensis* from the Upper Triassic of southern Brazil (photograph in (*f*) was published under creative commons attribution licence [13]). As pointed out by the arrows in (*a*), a number of bony structures were present when YPM VPPU 18835 was initially found, but have been subsequently lost. These include parts of 1 premaxillae, 2 right maxilla, 3 left jugal, 4 and 5 right jugal.

(figure 1*c*). Posterior to the facial process is a long and slender posterior process that extends posteriorly to just beyond the midpoint of the orbit (figure 1*c*). Immediately posterior to the facial process, the maxilla forms the ventral margin of a large lacrimal foramen. The posterior process of the maxilla has an L-shaped cross-section forming the medial articulation facet for the jugal (figure 1*c,d*). No teeth are visible in the left maxilla, which we attribute to the fact that much of the jawbones were lost during initial recovery. In addition, there are no alveolar structures visible on the ventral margin of the bone that would indicate thecodont or subthecodont tooth implementation.

The flat dorsal parts of the nasals are incompletely preserved (figure 1*a,b,d*). Each nasal has a large contact with the prefrontal posterolaterally, before forming a posterior lobe that overlaps a depressed articular surface on the anterior part of the frontals (e.g. cf. *Sphenodon* [10]). The lateral part of the nasal curves strongly ventrally (figure 1*c,d*), with the curvature resembling that seen in the prefrontal. As preserved, probably due to crushing of the specimen, the lateral part of each nasal is completely overlapped by the maxilla. When removing the maxillae in the virtual model (figure 1*g*), a maxillary facet is visible on each nasal, indicating that the maxilla would have overlapped only about half the dorsoventral height of the lateral surface of the nasal in life.

The right prefrontal is slightly more complete than the left one. The prefrontal has a broad contact with the nasal anteriorly and medially, and with the maxilla anterolaterally (figure 1*a,d,g*). Dorsally, the prefrontal fits into a deep articular facet on the lateral margin of the frontal (figure 1*e,f*). Ventrolaterally, the prefrontal forms the dorsal margin of a large lacrimal foramen. There is no evidence for a distinct lacrimal.

Much more of the left jugal is preserved compared to the right one (figure 1*a*), of which only the midportion articulating with the ectopterygoid and the ascending process is preserved. Prior to initial preparation, both jugals were more complete. The left bone shows that the jugal is massive and robust, and forms a broad contact with the prefrontal anteriorly and a long contact with the posterior process of the maxilla laterally. The jugal is expanded transversely at its mid-portion, below the orbit, in ventral view. From this mid-part, the ascending process of the jugal extends dorsally to articulate with the descending process of the postorbital, and a tapering posterior process extends posteroventrally (figure 1*c*). This posterior process probably ended freely (i.e. did not articulate with the quadrate or quadratojugal, if present, to form a closed lower temporal bar).

The frontals are paired (figure 1a). The anterior margin of the frontal possesses a concave lobate articular facet for the nasal that medially extends into a thin anterior frontal process. The tip of the process appears to be incomplete on both sides. Anterolaterally, the frontal has a deep facet for articulation with the prefrontal (figure 1e), and posterolaterally a similarly deep facet for the anterior process of the postfrontal. Based on the right side, the frontal was nearly excluded from the orbital rim by the contact between the pre- and postfrontals (figure 1f). Posteriorly, the posterior margin of each frontal is concave but with irregular processes that extend short distances into a large fontanelle between the frontals and the parietals (figure 1a).

The postfrontals are tripartite bones (figure 1a,e,f). Each has a tapering anterior process that articulates with the frontal and the prefrontal, a tapering ventral process that articulates with the postorbital, and a broader and short medial process that overlaps the parietal.

Both postorbitals are damaged, with the left one showing a better-preserved descending process (figure 1a). This descending process articulates with the ascending process of the jugal. The remainder of the postorbital is lappet-shaped, but only the medial part of the lappet is preserved, which would have articulated with the forked anterior process of the squamosal. Medially, a tapering process of the postorbital slightly overlaps the postfrontal dorsally (figure 1a).

The parietals are not fused to each other and have a visible midline suture (figure 1a). Their anterior parts are well preserved, whereas the posterolateral processes that would have articulated with the squamosals are not preserved. Anteriorly, the margins of the parietals are concave and appear incompletely ossified, similar to the posterior margins of the frontals, together with which they frame a large median fontanelle. Anteriorly, the position of the parietal foramen is clearly visible, despite the parietals being incomplete in this region. Posterior to the parietal foramen, the parietals meet medially to form a low sagittal crest.

Ventral to the posteromedial margin of the parietals several isolated bits of bone are interpreted as the probable remains of the supraoccipital (figure 1b). Ventral and posterior to the presumed supraoccipital, a slightly posteriorly convex bit of bone probably represents a partial basioccipital. An angled rod-like bone with an extended footplate that lies deep within the left upper temporal fenestra is interpreted as the left epipterygoid (figure 1b,c). This element has slightly shifted out of contact with the left pterygoid.

## 3.3. Palatal region of the skull

Of the palate (figure 1b), the left vomer, the anterior portions of both left and right palatines, a partial right ectopterygoid and incomplete portions of both pterygoids are present. As preserved, none of the elements shows clear evidence for the presence of teeth.

The left vomer is an elongated and transversely broad bone that tapers anteriorly (figure 1b). Because the element lies isolated anterior to the palatine and medially to the maxilla, its potential articulation with either of the bones remains unresolved. The posterolateral border is slightly concave in ventral view.

Only the anterior parts of the palatines are preserved, but they are strongly mediolaterally expanded (figure 1b). Anteriorly, the margins of the palatines are slightly concave in ventral view, and the bones extend anteromedially almost level with the tips of the anterior processes of the pterygoids.

The right pterygoid is represented by the long and tapering anterior process (figure 1b), most of the lateral flange and a thin remnant of the posterior quadrate flange. The lateral flange is still in articulation with the medial process of the incomplete right ectopterygoid (figure 1b). The left pterygoid has only the tapering anterior process and the posterior quadrate flange preserved. Based on these incomplete parts, the pterygoids had wide and flaring lateral flanges and similarly laterally expanded quadrate flanges extending from anteromedially to posterolaterally. The remnants of the quadrate flanges are also straight laterally and end in a small hook medially (figure 1b).

The incomplete right ectopterygoid articulates with the jugal laterally and with the anterior part of the preserved lateral flange of the right pterygoid medially (figure 1b). The preserved part appears slightly curved along its length and its contacts with adjacent bones are expanded anteroposteriorly. Although the shape of the bone is similar to the ectopterygoids of other rhynchocephalians, we cannot rule out, however, that the curved central part of the bone might have been a bit wider in life.

## 3.4. Lower jaw

Of the mandibular rami, only the massive and convex coronoid processes of the dentaries are present, with the right one preserving more of the posterior margin than the left one (figure 1a–c). The flat medial surface of the right coronoid process is also angled, mirroring the inclination of the adjacent lateral margin of the

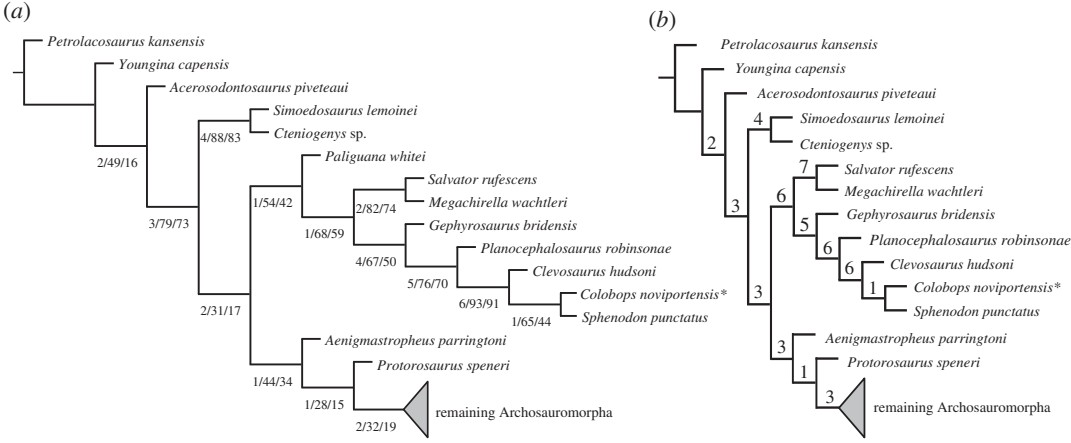

**Figure 3.** Phylogenetic framework of the present study showing the position of *Colobops noviportensis* (highlighted by an asterisk) within Rhynchocephalia (note that the trees have been collapsed at the node 'remaining Archosauromorpha'; for the full trees, see electronic supplementary material, figures S2 and S3). (*a*) Analysis including *Paliguana whitei*. The numbers below lines indicate decay indices and bootstrap values (absolute and GC; see Material and methods). (*b*) Analysis with *P. whitei* being pruned *a posteriori* from the dataset. Numbers above lines indicate decay indices only.

pterygoid flange. The posterior sloping part of the right coronoid process indicates an extended contact with the surangular (comparable to the condition of *Clevosaurus cambrica* [12], their figure 10).

## 3.5. Skull openings

Several skull openings can be reconstructed for YPM VPPU 18835 (figure 1*a,b*), including the orbits and most of the supratemporal fenestrae and a ventrally opened infratemporal fenestra (= as a morphological ventral emargination), as well as some openings of the palate. The orbits are framed by the prefrontals anteriorly and anterodorsally, the postfrontals dorsally and posterodorsally, the postorbitals posteriorly and the jugals posteroventrally and ventrally. An open ventral emargination instead of a closed lower temporal arcade is probably present, based on the morphology of the jugal (figure 1*c*). The ventral emargination is delimited anteriorly and anteroventrally by the jugal, dorsally by the bar formed by the postorbital and squamosal, and posteriorly it would have been framed by the quadrate (and potentially the quadratojugal if present). The supratemporal fenestra is framed by the postfrontal anteriorly, the parietal medially and probably posteriorly, by the squamosal posterolaterally and laterally, and by the postorbital anterolaterally. On the palate, a small internal narial opening was framed by the vomer anteromedially, the maxilla anterolaterally and the palatine posteriorly. Due to the lack of preservation in other palatal regions, the outline and bones framing the suborbital fenestra could not be reconstructed. The two coronoid processes are still preserved within the subtemporal fenestra. The anterior margin of the subtemporal fenestra is formed by the ectopterygoid and medially by the pterygoid flange, whereas the lateral margin is comprised by the jugal.

## 3.6. Results of new phylogenetic analyses

The first analysis using the modified version of the matrix provided by Pritchard *et al.* [6] recovered nine most parsimonious trees (MPTs), each with a tree length of 1099 steps (one step shorter than in the original analysis) and a consistency index of 0.3267 and a retention index of 0.6459. The topology of the strict consensus tree has a massive polytomy composed of the main clades of Sauria (electronic supplementary material, figure S2). This lack of resolution is mainly a result of the two alternative positions that *Colobops noviportensis* acquires among the MPTs, either as a lepidosauromorph or the earliest diverging rhynchosaur. The second analysis of the modified version of the data matrix of Ezcurra [17] (as modified by subsequent authors; see Material and methods) found 27 MPTs with a tree length of 3864 steps and a consistency index of 0.2399 and a retention index of 0.6461. The topology of the strict consensus tree (figure 3; electronic supplementary material, figure S3) is completely congruent with that recovered by other recent analyses of this dataset [18,19]. *Colobops noviportensis* was found among Lepidosauromorpha and Rhynchocephalia in each MPT, contrasting

with its placement within Archosauromorpha and Rhynchosauria [6]. Within Rhynchocephalia, *Colobops noviportensis* was recovered as the sister taxon to the only extant rhynchocephalian, *Sphenodon punctatus*. Among the sampled lepidosauromorphs, *Paliguana whitei*, *Salvator rufescens* + *Megachirella wachtleri*, *Gephyrosaurus bridensis*, *Planocephalosaurus robinsonae* and *Clevosaurus hudsoni* represent the successive sister taxa to the *Colobops noviportensis* + *Sphenodon punctatus* clade.

The recovered branch supports are relatively low at Lepidosauromorpha and Lepidosauria (Bremer supports of 1 and bootstrap frequencies lower than 70%). These low supports at these branches are very likely a result of ambiguous optimizations produced by the poorly known *Paliguana whitei* (figure 3*a*). Indeed, if this species is pruned *a posteriori* (figure 3*b*, electronic supplementary material, figure S4), the Bremer supports of Lepidosauromorpha and Lepidosauria increase to six. By contrast, the branch supports are considerably higher within Rhynchocephalia (Bremer supports of 4 or higher) and in particular, the branch that includes *Clevosaurus hudsoni* and *Colobops noviportensis* + *Sphenodon punctatus*, with a Bremer value of 6 and absolute and GC bootstrap frequencies of 93% and 91%, respectively. Nevertheless, the levels of support for the *Colobops noviportensis* + *Sphenodon punctatus* branch are the lowest within Rhynchocephalia and this is probably a combination of the incomplete and damaged condition of the only known specimen of *Colobops noviportensis* and conflicting evidence among the sample of eusphenodontian rhynchocephalians included in this dataset. The latter is suggested by a difference of 21% between the absolute and GC bootstrap frequencies in this branch. However, the aim of this dataset was not to assess the position of *Colobops* among rhynchocephalians more crownward than *Planocephalosaurus*.

Suboptimal searches constraining the position of *Colobops noviportensis* in different positions of the tree found that nine additional steps are necessary to force its position as the most basal rhynchocephalian lepidosauromorph (forcing the monophyly of all other rhynchocephalians), 12 to be placed as the most basal lepidosauromorph (forcing the monophyly of all other lepidosauromorphs), 13 to be placed as an archosauromorph (found as one of the sister taxa to Tanystropheidae after forcing the monophyly of *Colobops* + archosauromorphs), and 17 to be the earliest diverging member of Rhynchosauria (forcing the monophyly of *Colobops* + rhynchosaurs, resembling the phylogenetic placement of this species found by Prichard *et al.* [6]). As a result, the position of *Colobops* as an archosauromorph and, in particular, as a rhynchosaur is highly unlikely based on the current phylogenetic dataset, whereas its placement as a rhynchocephalian, as originally suggested [5] is strongly supported.

## 3.7. Characterization of adductor development

Depending on how the posterior process of the postorbital is reconstructed in the skull of *Colobops noviportensis* (figure 4), the articulation with the squamosal and the shape and size of the supratemporal fenestra varies to some degree (table 1). When correcting for post-mortem compression of the skull, the squamosal needs to be shifted medially for it to be in position to articulate with the posterior process of the postorbital (figure 4*b*,*c*). This leads to an overall narrower supratemporal fenestra and space for the adductor muscle attachment (figure 5).

# 4. Discussion

## 4.1. Rhynchosaurian versus rhynchocephalian affinities

Pritchard *et al.* [6] (their fig. 5) cited four features that they considered unambiguous synapomorphies for *Colobops* and Rhynchosauria: (i) anterolateral lamina of the maxilla overlapping a posterolateral process of the premaxilla, (ii) rostral length less than 40 per cent of the total skull length, (iii) ventrolateral lamina of the nasal laps medial to the dorsal process of the maxilla, and (iv) upper temporal bar continuous with the dorsal margin of the orbit. All four features are also present in a number of rhynchocephalians including the Late Triassic to Early Jurassic *Clevosaurus* [26,27] (figure 2*f*) and the extant *Sphenodon* [10,28]. Accordingly, we recovered *Colobops* deeply nested within rhynchocephalians (figure 3), sharing with *Sphenodon* the presence of a robust bony connection between the prefrontal and palatine [10] and with *Clevosaurus* and *Sphenodon* an anteroposteriorly short snout (30% or less of total skull length). Of the features listed as diagnostic for *Colobops* [6], the near-exclusion of the frontal from the orbital rim due to the contact between prefrontal and postfrontal is also found in *Clevosaurus* [27] and *Sphenodon* [10], although to a lesser degree. The 'posteriorly directed process near the transverse midpoint of the supratemporal margin' probably

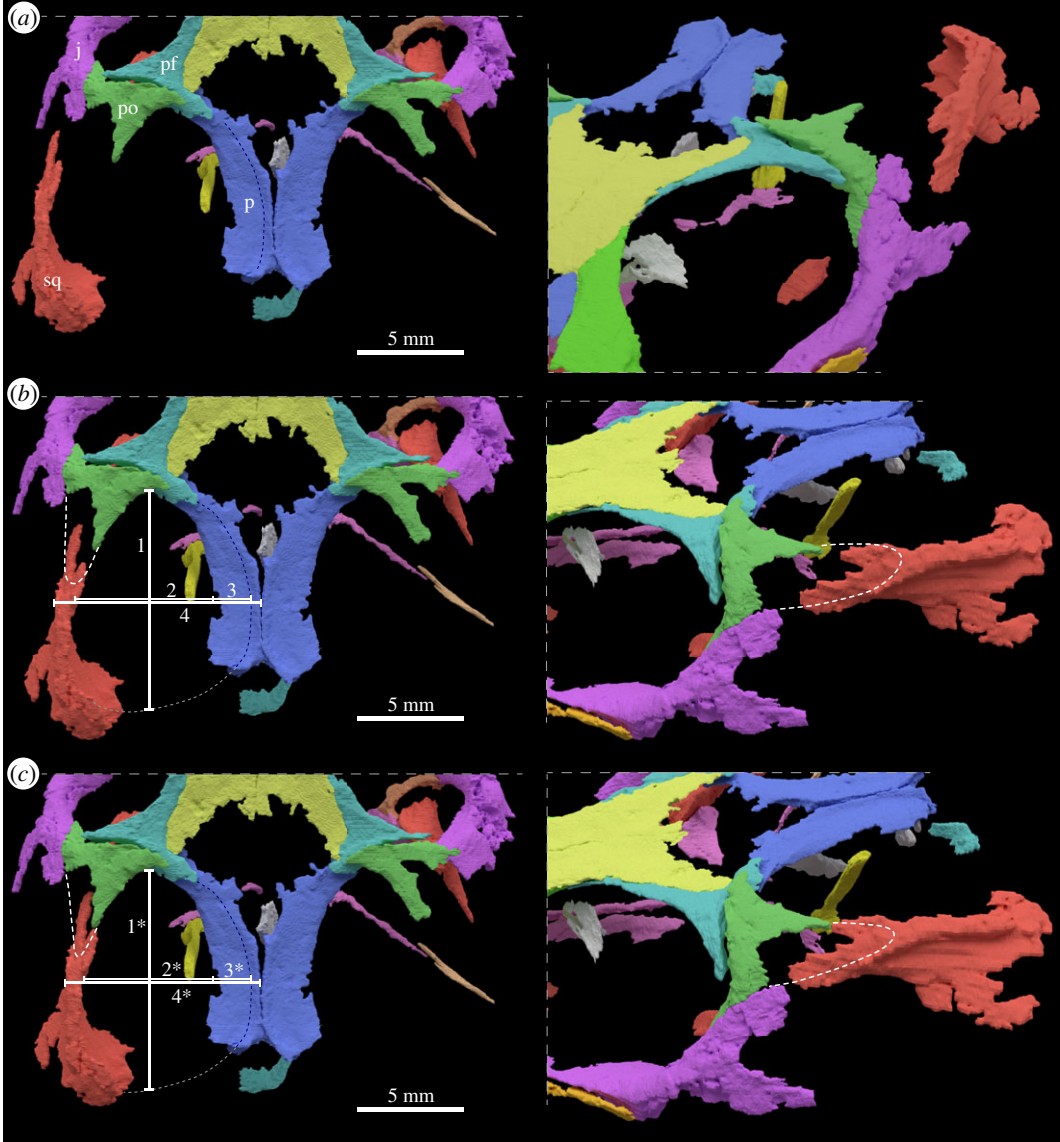

**Figure 4.** Close-up of the supratemporal region of YPM VPPU 18835. Images in (*a*) show the bones framing the supratemporal fenestra as preserved in dorsal and angled anterolaterodorsal view. Images in (*b*,*c*) represent alternative configurations based on different reconstructions of the posterior lappet-shape of the postorbital (indicated by a white stippled line on the lateral surface of the squamosal, marked in red) and its 'in life' position, contacting the squamosal. The medial and posterior extent and shape of the supratemporal fenestra is indicated by a stippled blue and white line. j, jugal; p, parietal; pf, postfrontal; po, postorbital; sq, squamosal. Numbered white lines indicate measurements provided in table 1.

represents the incomplete posterior portion of the postorbital, which forms a broad lappet in the upper temporal bar of *Clevosaurus* and closely related rhynchocephalians [26]. The shape of the broad posterior lappet is reflected by the large articular surface on the anterior process of the squamosal. Accordingly, there is no free posterior process of the postorbital extending into the supratemporal fenestra. This suggests that the upper temporal openings of *Colobops* were not unusually large (contra [6]). Instead, based on our revised measurements, *Colobops* falls well within the prediction intervals, and in a general space occupied, among other groups, by other rhynchocephalians in the dataset (figure 5*a*,*b*).

The anterior concavity of the palatine between the lateral contact with the maxilla and anteromedial contact with the vomer in *Colobops*, defining the posterior margin of the choana, is also transversely broad in rhynchocephalians. Furthermore, the medial process of the postorbital overlaps the postfrontal and does not reach the parietal. The medial process has a tapering end and is directed posteromedially, and is thus similar to *Clevosaurus bairdi* [27] but not to *C. brasiliensis*, where it is directed anteromedially [29], or *C. hudsoni*, where it is squared-off [26]. In addition, *Colobops* differs

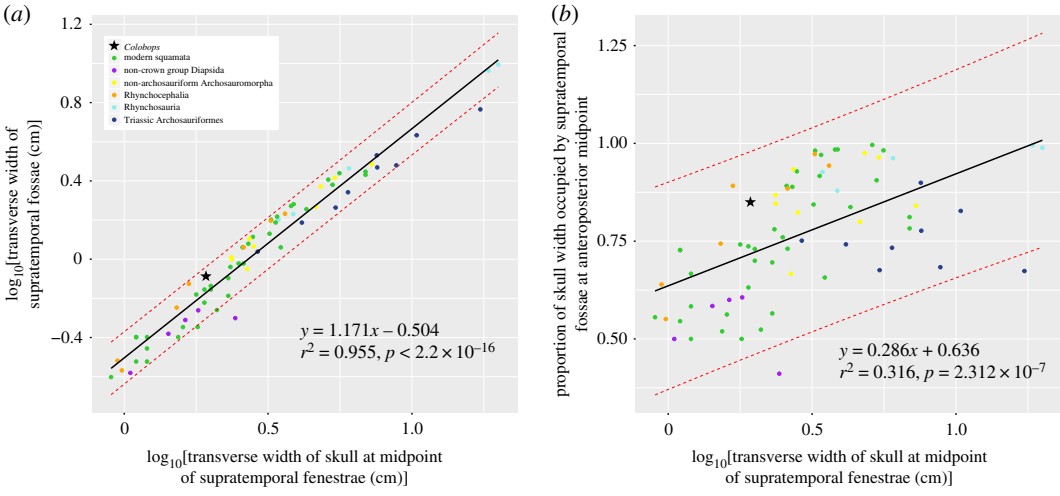

**Figure 5.** Bivariate plots of overall skull size against supratemporal fossa dimensions. (*a*) Log$_{10}$-transformed bivariate plot of transverse skull width against transverse supratemporal fossa width. (*b*) Log$_{10}$-transformed bivariate plot of transverse skull width against proportion of skull width occupied by supratemporal fossa at anteroposterior midpoint. The solid line indicates the results of the linear regression, and dashed lines indicate confidence intervals (90% for *a*, 95% for *b*).

from non-rhynchosaurid rhynchosaurians (*Mesosuchus browni*, *Howesia browni* and *Eohyosaurus wolvaardti*) in the presence of the following features: sagittal crest posterior to the parietal foramen, postorbital overlapping the postfrontal dorsally, posterior end of the maxilla extensively overlapping laterally the jugal, acute and posterolaterally oriented apex of the lateral flange of the pterygoid, coronoid process more dorsally developed, transversely broad palatines, thin and slit-like palatal ramus of the pterygoid, absence of ornamentation on the skull roof, contact between the postorbital and parietal, contact between the maxilla and ectopterygoid, contact between the palatal rami of the pterygoids, and posterior hook on the lateral ramus of the ectopterygoid.

## 4.2. Functional implications

The argument that *Colobops noviportensis* had an exceptionally strong bite [6] was largely based on the assumption that most of the bones are preserved in more or less natural anatomical position. Following this interpretation, all measurements regarding the total width of the skull or the supratemporal fenestra in particular were then used to infer functional capabilities.

Pritchard *et al*. [6] commented on the difficult task of separating the rostral bones from the surrounding rock matrix that often shows similar densities (i.e. grey values) to the bone in the scan [7]. We cannot, however, corroborate the configuration of the anterior skull bones in *Colobops noviportensis* including the 'stacking' of premaxilla, maxilla and nasal bones as presented by [6]. We found what these authors interpreted as a part of the premaxilla to be continuous with the maxilla (figure 1*a–d*); their interpretation [6] is further not supported by old photographs that show that the maxilla was much wider and extended a bit further anteriorly prior to the initial preparation phase (figure 2). As such, only the maxillae and the nasals are still present in the snout region, with much of the overlapping of these two cranial bones being due to taphonomic compaction of the rostrum.

We furthermore interpret the entire skull as being severely dorsoventrally flattened (figure 1*c*), which caused strong displacement of most skull bones aside from those forming the skull roof. This post-mortem distortion is expressed by (i) the lateral angling of the maxilla and the jugal, (ii) the complete overlap of the lateral border of the nasal by the maxilla, (iii) the increased space between jugal and the coronoid bone (as part of the lower jaw), (iv) the strong lateral displacement of the left squamosal compared to the remaining part of the left postorbital, (v) the strongly oblong orbits, (vi) the posterior tilt of the epipterygoid columella, (vii) the separation and tilting of the postorbital relative to the postfrontal, and (viii) the slight separation of the parietals along the sagittal crest directly posterior to the parietal foramen. As such, any measurement of the skull width and height, as well as that of any cranial opening (orbits, temporal fenestrae) directly on the fossil or 3D rendering are not reliable. If the skull is reconstructed deeper and less flattened, the orbits would be more circular and the supratemporal fenestra would have been much narrower than previously reconstructed [6]. The

squamosal must be shifted into its natural position, in which the forked anterior process articulates with the originally lappet-like posterior process of the postorbital. The forked anterior process of the squamosal carries a clear depression representing the postorbital facet (figure 4).

Why was the squamosal assumed previously [6] to be in natural position in the scan data (figure 4*a*)? We infer this to be linked to several aspects, namely an uncertainty of what exactly is the shape of the jugal and misinterpretation of the jugal–postorbital contact, which leads to a much larger postorbital and its dorsal exposure than is actually present (see Pritchard *et al.* [6] character 4 of the diagnosis of *Colobops noviportensis*). Linked to this issue is the misinterpretation of the width of the upper temporal bar (not the 'postorbital bar' as is mentioned in the figure caption in their electronic supplementary material, figure 13) formed by the postorbital and the squamosal (their character 315). These misinterpretations seem to reflect mismatching coloration of the segmented skull bones: in their fig. 1*a*, the posterior part of the left jugal is coloured green, i.e. the same colour as the postorbital (not a pale lavender as the anterior part of the jugal), whereas in fig. 1*b–d* the posterior part of the jugal is coloured a light grey, which according to the authors, was used for skull parts of questionable homology.

The shifting and loss of contact of most cranial bones is also in agreement with a more juvenile ontogenetic stage of *Colobops noviportensis*, in which the bones were less integrated and less firmly sutured with each other compared to the condition in adults (e.g. [10]). The presence of a large and wide fontanelle between the frontals and the parietals, and encompassing the parietal foramen, would thus be indeed a juvenile feature rather than a phylogenetically informative character. A similar configuration was reported in juvenile specimens of *Sphenodon* [30,31], and the shape of the fontanelle is also very similar to a hatchling specimen of *Sphenodon punctatus* (see electronic supplementary material, figure S1). Furthermore, embryonic and hatchling skulls of extant saurians do not show any indication during development in which the posterior process of the postorbital reaches freely halfway into the supratemporal fenestra, which would already house the developing adductor muscles. The posteriorly directed process of the postorbital into the supratemporal fenestra as interpreted by Pritchard *et al.* [6] is an artefact resulting from the damage to both postorbitals. Comparison with a variety of rhynchocephalians [26,27] indicates that this process is actually an incomplete posterior process of the postorbital, which overlaps the squamosal on the upper temporal bar. In addition, the postorbital develops in close contact with the anterior process of the squamosal [32–34]. Based on the comparison with other extinct and extant saurians, it is apparent that the squamosal was clearly displaced during burial and fossilization of YPM VPPU 18835.

Depending on how the posterior process of the postorbital is reconstructed (figure 4*b,c*) the articulation with the squamosal and the shape and size of the supratemporal fenestra vary to some degree (table 1). In the first reconstruction (figure 4*b*), the postorbital is reconstructed with a wider lappet-like posterior process, whereas the process is reconstructed thinner and more tapering posteriorly in the second reconstruction (figure 4*c*). In the first case, the squamosal is shifted less medially to articulate with the posterior process of the postorbital, whereas it is shifted more medially in the latter case. Both reconstructions, however, indicate that the supratemporal fenestra was less mediolaterally expanded, leading to a lower 'adductor chamber/total width' ratio than was previously reported. The new ratio (between 0.85 and 0.86; see also figure 5) and surface area measurements of the supratemporal fenestra lie within the variation observable in extant and extinct reptiles ([6] their electronic supplementary material, table S1). *Colobops noviportensis* is not an outlier in either regression (figure 5), indicating that its supratemporal fossae are not unusually large for its size, and instead plots in a similar position to other rhynchocephalians.

## 5. Conclusion

To conclude, even though *Colobops noviportensis* is not a hyper-specialized rhynchosaur that yields insights into early diapsid feeding apparatus evolution, it was nevertheless one of many small diapsids known from the Triassic that together are part of a reasonably well-documented diversity of cranial anatomy and associated feeding apparatus [35–38].

Ethics. Preparation and scanning of specimens used herein was done for previous studies and as such, this point does not apply to the present paper.

Data accessibility. All data generated by the present study are enclosed either in the main article or in the accompanying electronic supplementary material, files. The original CT dataset of *Colobops noviportensis* is linked to [6] and available upon request from the corresponding authors of that study.

Authors' contributions. T.M.S., S.N.F.S., R.J.B. and H.-D.S. conceived of the study. T.M.S., H.-D.S. and S.N.F.S. wrote the initial manuscript. S.N.F.S. and T.M.S. performed the virtual reconstruction. M.D.E. and M.E.H.J. performed phylogenetic analyses and provided insights into character evolution, and R.J.B. performed the regression analyses of the upper temporal region. All authors contributed to the discussion and finalization of the manuscript.
Competing interests. We declare we have no competing interests.
Funding. T.M.S. and S.N.F.S. acknowledge support by the Swiss National Science Foundation (grant no. 205321_162775).
Acknowledgements. We are greatly indebted to A. Pritchard, B.-A. Bhullar and colleagues (YPM), as well as to the Center for Nanoscale Systems at Harvard University (Cambridge, MA) for making the *Colobops* CT scan freely available for study. G. Sobral (SMNS) and J. Müller (MFN) and C. Schultz (UFRGS) are thanked for providing the scans for *Mesosuchus* and *Teyumbaita*, respectively. C. Bell and the UTCT are acknowledged for making the scan of the juvenile *Sphenodo*n freely available. The program TNT is kindly made available through the sponsorship of the Willi Hennig Society. Finally, we like to thank A. Dunn and the editorial team, reviewer G. Sobral and one anonymous reviewer for their constructive comments on the previous manuscript draft.

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
