## [Reviewer comments · Royal Society Open Science]

***Colobops*: a juvenile rhynchocephalian reptile
(Lepidosauromorpha), not a diminutive archosauromorph
with an unusually strong bite**

Torsten M. Scheyer, Stephan N. F. Spiekman, Hans-Dieter Sues, Martín D. Ezcurra,
Richard J. Butler and Marc. E. H. Jones

Article citation details

R. Soc. open sci. **7**: 192179.
<http://dx.doi.org/10.1098/rsos.192179>

Review timeline

Original submission: 13 December 2019
Revised submission: 26 February 2020
Final acceptance: 27 February 2020

Note: Reports are unedited and appear as
submitted by the referee. The review history
appears in chronological order.

Review History

RSOS-192179.R0 (Original submission)

Review form: Reviewer 1 (Gabriela Sobral)

Is the manuscript scientifically sound in its present form?

Yes

Are the interpretations and conclusions justified by the results?

Yes

Is the language acceptable?

Yes

Do you have any ethical concerns with this paper?

No

Have you any concerns about statistical analyses in this paper?

No

Recommendation?

Accept with minor revision (please list in comments)

Comments to the Author(s)

Dear authors,

Congratulations on your manuscript on the anatomical reinterpretation of Colobops. The manuscript is well-written and the argument supporting this new hypothesis is very sound. The manuscript is well illustrated and the analyses are robust.

The current manuscript adds new, relevant information to the body of literature on Triassic lepidosauromorphs. It also corrects fundamental problems with the more recent publication on the taxon, raising several important issues that palaeontologists should be aware of when interpreting anatomical data.

I have only minor suggestions and they are found below.
I thus recommend the manuscript be accepted with minor revisions.

Gabriela Sobral

--

In general, the authors could have made better use of figure callings, especially indicating more precisely where the information is to be found (eg.: 1B, 2D etc). I found it very difficult to indicate where my comments were referring to and I suggest next time the authors use a continuous numbering of lines, instead of starting anew every page.

Page 4

Line 11: "and for two of those they indicated that the character state might reflect immaturity." This sentence reads a bit strange.. perhaps: "two of which indicated as possibly reflecting immaturity"

Line 35: "images of raw data". 3D models may be interpreted as raw data as a prepared fossil; I think the authors may want to stress that it's the segmented model that represents an anatomical hypothesis.

Page 5

Line 22: "is" instead of "are"

Line 36: " $(2+3) \times 2/4$ " I would suggest to change to letters or roman numerals, at first I thought I had to do some calculations here.

Page 7

In general, I think the authors need to explicit to the reader why they are using a new data matrix for their phylogenetic analysis instead of the one provided by the original study. The same for the addition of new characters and new taxa, and modification of previous characters: you need to explain why you made this decision. Although I know this is not the case, it could be argued that this is data manipulation with the sole purpose of contradicting the original analysis.

Another thing is that there is a recently published dataset for rhynchocephalians, which could be used for a more precise analysis within the group. Maybe the authors would like to take the chance and consider adding a second phylogenetic analysis: Hsiou et al. 2019. A New Clevosaurid from the Triassic (Carnian) of Brazil and the Rise of Sphenodontians in Gondwana (Scientific Reports).

Page 9

Line 14: braincase elements not preserved. From figure 1B, the gray element marked as "?" looks like the postero-lateral portion of the basisphenoid that contributes to the basal tubera, no?

Page 12

Palatal section: a (perhaps supplementary) figure showing a palatal view of the skull without the bones of the skull roof would be helpful.

Page 13

Line 32: I don't understand what exactly is an artefact. It seems to perfectly match the palatal anatomy of other rhynchocephalians.

Lower jaw: a figure of the lower jaw is missing. It could come in the SuppMat, but I would suggest to put it in the main text since some features of the coronoid eminence are discussed elsewhere in the text.

Page 14

Line 6: I would avoid the term "ventral emargination". To my knowledge, it is restricted to the ichthyosaur/sauropterygian literature and has a somewhat different anatomical context. I would keep the "ventrally open lower temporal fenestra".

Skull openings: Since the authors took time to write about them, I would also include the subtemporal fenestra. For instance, it is interesting to notice the ectopterygoid also participates in it... it's not always the case

Page 15

Line 15: sister taxa instead of closer taxa

Line 27: Why prune Paliguana a posteriori to test Bremer supports, instead of a priori?

Page 19

Adductor development: first, I would open this section stressing to the reader that the posterior process is incomplete. Second, I would explore more the differences in interpretations for the position of the squamosal, otherwise figure 3 feels mostly unnecessary. So either a good part of figure 3 could be moved to the SuppMat, or the discussion of these differences expanded.

Page 18

Line 22: add "in the latter" after "following features" - as it stands, it is confusing if the features discussed refer to Colobops or rhynchosaurus

Line 29: Instead of "lateral process", the authors used "lateral flange" in the description. I would keep consistency.

Lines 31 and 38: is it really possible to state that the features regarding the palatal rami of the pterygoids can be assessed for sure in Colobops? It didn't feel so by the description and images.

Page 19

Line 43: Why is the left quadrate not segmented or shown in the figures? Or did the authors mean the right quadrate (and right squamosal)?

Page 21

Lines 25 and 26: the following sentence reads weird in relation to the previous one: "In contrast, the postorbital develops in close contact with the anterior process of the squamosal".

Line 34 and 35: What is the difference between the two reconstructions? Please make it explicit.

FIGURES

Fig.1: This figure looks great, but it is a bit confusing. I am not also sure there is a need for 1D-G to be shown in the main text; they could be maybe moved to the SuppMat.

Fig.3: Please consider to somehow highlight the name of the taxon in the topology.

Fig.4: Regardless of being discussed or not in further depth in the "Adductor Development" section, this figure could be reduced and have some views moved into the SuppMat (esp. A). Also, I would invert the left and the right columns to show first the putative position of the squamosal and second the measurements.

Review form: Reviewer 2 (Annie Hsiou)

Is the manuscript scientifically sound in its present form?

Yes

Are the interpretations and conclusions justified by the results?

Yes

Is the language acceptable?

Yes

Do you have any ethical concerns with this paper?

Yes

Have you any concerns about statistical analyses in this paper?

No

Recommendation?

Accept with minor revision (please list in comments)

Comments to the Author(s)

My only concern is about the systematic position of *Colobops* among the Sphenodontia. Could the authors include a more comprehensive phylogenetic analysis with the phylogenetic hypotheses available for Sphenodontia to compare with the actual phylogenetic data of the manuscript? This would help to try to clarify *Colobops*' closest relationships with other sphenodontian. However, it is not a closed condition, although I believe among the other sphenodontian researchers, this is a primary condition and very important to understanding the Rhynchocephalia evolution.

Decision letter (RSOS-192179.R0)

18-Feb-2020

Dear Dr Scheyer

On behalf of the Editors, I am pleased to inform you that your Manuscript RSOS-192179 entitled "*Colobops*: a juvenile rhynchocephalian reptile (Lepidosauromorpha), not a diminutive archosauromorph with an unusually strong bite" has been accepted for publication in Royal Society Open Science subject to minor revision in accordance with the referee suggestions. Please find the referees' comments at the end of this email.

The reviewers and handling editors have recommended publication, but also suggest some minor revisions to your manuscript. Therefore, I invite you to respond to the comments and revise your manuscript.

- Ethics statement

- Data accessibility

<http://datadryad.org/submit?journalID=RSOS&manu=RSOS-192179>

- Competing interests

- Authors' contributions

- Acknowledgements

- Funding statement

Because the schedule for publication is very tight, it is a condition of publication that you submit the revised version of your manuscript before 27-Feb-2020. Please note that the revision deadline will expire at 00.00am on this date. If you do not think you will be able to meet this date please let me know immediately.

To revise your manuscript, log into <https://mc.manuscriptcentral.com/rsos> and enter your Author Centre, where you will find your manuscript title listed under "Manuscripts with

Decisions". Under "Actions," click on "Create a Revision." You will be unable to make your revisions on the originally submitted version of the manuscript. Instead, revise your manuscript and upload a new version through your Author Centre.

If your manuscript is newly submitted and subsequently accepted for publication, you will be asked to pay the article processing charge, unless you request a waiver and this is approved by Royal Society Publishing. You can find out more about the charges at <https://royalsocietypublishing.org/rsos/charges>. Should you have any queries, please contact opscience@royalsociety.org.

on behalf of Professor Marcelo Sanchez (Associate Editor) and Jon Blundy (Subject Editor)
 openscience@royalsociety.org

Reviewer comments to Author:

Reviewer: 1

Comments to the Author(s)

Dear authors,

Congratulations on your manuscript on the anatomical reinterpretation of Colobops. The manuscript is well-written and the argument supporting this new hypothesis is very sound. The manuscript is well illustrated and the analyses are robust.

The current manuscript adds new, relevant information to the body of literature on Triassic lepidosauromorphs. It also corrects fundamental problems with the more recent publication on the taxon, raising several important issues that palaeontologists should be aware of when interpreting anatomical data.

I have only minor suggestions and they are found below.
 I thus recommend the manuscript be accepted with minor revisions.

Gabriela Sobral

--

In general, the authors could have made better use of figure callings, especially indicating more precisely where the information is to be found (eg.: 1B, 2D etc). I found it very difficult to indicate where my comments were referring to and I suggest next time the authors use a continuous numbering of lines, instead of starting anew every page.

Page 4

Line 11: "and for two of those they indicated that the character state might reflect immaturity."
 This sentence reads a bit strange.. perhaps: "two of which indicated as possibly reflecting immaturity"

Line 35: "images of raw data". 3D models may be interpreted as raw data as a prepared fossil; I think the authors may want to stress that it's the segmented model that represents an anatomical hypothesis.

Page 5

Line 22: "is" instead of "are"

Line 36: "(2+3)x2/4" I would suggest to change to letters or roman numerals, at first I thought I had to do some calculations here.

Page 7

In general, I think the authors need to explicit to the reader why they are using a new data matrix for their phylogenetic analysis instead of the one provided by the original study. The same for the

addition of new characters and new taxa, and modification of previous characters: you need to explain why you made this decision. Although I know this is not the case, it could be argued that this is data manipulation with the sole purpose of contradicting the original analysis.

Another thing is that there is a recently published dataset for rhynchocephalians, which could be used for a more precise analysis within the group. Maybe the authors would like to take the chance and consider adding a second phylogenetic analysis: Hsiou et al. 2019. A New Clevosaurid from the Triassic (Carnian) of Brazil and the Rise of Sphenodontians in Gondwana (Scientific Reports).

Page 9

Line 14: braincase elements not preserved. From figure 1B, the gray element marked as “?” looks like the postero-lateral portion of the basisphenoid that contributes to the basal tubera, no?

Page 12

Palatal section: a (perhaps supplementary) figure showing a palatal view of the skull without the bones of the skull roof would be helpful.

Page 13

Line 32: I don't understand what exactly is an artefact. It seems to perfectly match the palatal anatomy of other rhynchocephalians.

Lower jaw: a figure of the lower jaw is missing. It could come in the SuppMat, but I would suggest to put it in the main text since some features of the coronoid eminence are discussed elsewhere in the text.

Page 14

Line 6: I would avoid the term “ventral emargination”. To my knowledge, it is restricted to the ichthyosaur/sauropterygian literature and has a somewhat different anatomical context. I would keep the “ventrally open lower temporal fenestra”.

Skull openings: Since the authors took time to write about them, I would also include the subtemporal fenestra. For instance, it is interesting to notice the ectopterygoid also participates in it... it's not always the case

Page 15

Line 15: sister taxa instead of closer taxa

Line 27: Why prune Paliguana a posteriori to test Bremer supports, instead of a priori?

Page 19

Adductor development: first, I would open this section stressing to the reader that the posterior process is incomplete. Second, I would explore more the differences in interpretations for the position of the squamosal, otherwise figure 3 feels mostly unnecessary. So either a good part of figure 3 could be moved to the SuppMat, or the discussion of these differences expanded.

Page 18

Line 22: add “in the latter” after “following features” – as it stands, it is confusing if the features discussed refer to Colobops or rhynchosaurs

Line 29: Instead of “lateral process”, the authors used “lateral flange” in the description. I would keep consistency.

Lines 31 and 38: is it really possible to state that the features regarding the palatal rami of the pterygoids can be assessed for sure in Colobops? It didn't feel so by the description and images.

Page 19

Line 43: Why is the left quadrate not segmented or shown in the figures? Or did the authors mean the right quadrate (and right squamosal)?

Page 21

Lines 25 and 26: the following sentence reads weird in relation to the previous one: "In contrast, the postorbital develops in close contact with the anterior process of the squamosal".

Line 34 and 35: What is the difference between the two reconstructions? Please make it explicit.

FIGURES

Fig.1: This figure looks great, but it is a bit confusing. I am not also sure there is a need for 1D-G to be shown in the main text; they could be maybe moved to the SuppMat.

Fig.3: Please consider to somehow highlight the name of the taxon in the topology.

Fig.4: Regardless of being discussed or not in further depth in the "Adductor Development" section, this figure could be reduced and have some views moved into the SuppMat (esp. A). Also, I would invert the left and the right collumns to show first the putative position of the squamosal and second the measurements.

Reviewer: 2

Comments to the Author(s)

My only concern is about the systematic position of *Colobops* among the Sphenodontia. Could the authors include a more comprehensive phylogenetic analysis with the phylogenetic hypotheses available for Sphenodontia to compare with the actual phylogenetic data of the manuscript? This would help to try to clarify *Colobops*' closest relationships with other sphenodontian. However, it is not a closed condition, although I believe among the other sphenodontian researchers, this is a primary condition and very important to understanding the Rhynchocephalia evolution.

Author's Response to Decision Letter for (RSOS-192179.R0)

See Appendix A.

Decision letter (RSOS-192179.R1)

27-Feb-2020

Dear Dr Scheyer,

It is a pleasure to accept your manuscript entitled "*Colobops*: a juvenile rhynchocephalian reptile (Lepidosauromorpha), not a diminutive archosauromorph with an unusually strong bite" in its current form for publication in Royal Society Open Science.

Due to rapid publication and an extremely tight schedule, if comments are not received, your paper may experience a delay in publication. Royal Society Open Science operates under a continuous publication model. Your article will be published straight into the next open issue and

this will be the final version of the paper. As such, it can be cited immediately by other researchers. As the issue version of your paper will be the only version to be published I would advise you to check your proofs thoroughly as changes cannot be made once the paper is published.

on behalf of Professor Marcelo Sanchez (Associate Editor) and Jon Blundy (Subject Editor)
openscience@royalsociety.org

Appendix A

Dear Dr Dunn,

Please find uploaded a revised version of our manuscript as including a tracked-changes version and a version where all changes have been included but not highlighted, as well as a detailed list of answers to the reviewers' suggestions/comments. We would like to thank the reviewers for their constructive comments.

I hope you find the introduced changes to your satisfaction.

With kind regards,

Torsten Scheyer (corresponding author)

Reviewers comments are numbered, black, and preceded by ###

Our reponses are numbered, blue, and preceded by >>>

Reviewer 1

(Gabriela Sobral):

1 ### In general, the authors could have made better use of figure callings, especially indicating more precisely where the information is to be found (eg.: 1B, 2D etc). I found it very difficult to indicate where my comments were referring to and I suggest next time the authors use a continuous numbering of lines, instead of starting anew every page.

1 >>> We have added more detailed references to the figures. The numbering system was, as we recall, added during the uploading process and is thus not in the hands of the authors to change.

Page4

2 ### Line 11: "and for two of those they indicated that the character state might reflect immaturity." This sentence reads a bit strange.. perhaps: "two of which indicated as possibly reflecting immaturity"

2 >>> In the first part of the sentence we talk of characters, not character states; therefore we cannot use "two of which" as has been proposed by the reviewer. We have slightly rewritten the sentence to increase readability: "The authors [6] mentioned five characters as diagnostic for *Colobops*

noviportensis. For two of those characters the authors indicated that the given state might reflect immaturity”

3 ### Line 35: “images of raw data”. 3D models may be interpreted as raw data as a prepared fossil; I think the authors may want to stress that it's the segmented model that represents an anatomical hypothesis.

3 >>> We do not agree with the reviewer in this point, because every virtual segmentation and creation of a model is subjective. We thus refrain from making changes to the text here. The fact that preparation of fossils is indeed subjective – to some degree – does not disprove our statement. We did add “virtual” to the text for clarification.

Page 5

4 ### Line 22: “is” instead of “are”

4 >>> Apologies, we meant “3d models” and have changed the text accordingly.

5 ### Line 36: “(2+3)x2/4” I would suggest to change to letters or roman numerals, at first I thought I had to do some calculations here.

5 >>> We have changed the text and the entry in the table to make it clearer.

Page 7

6 ### In general, I think the authors need to explicit to the reader why they are using a new data matrix for their phylogenetic analysis instead of the one provided by the original study. The same for the addition of new characters and new taxa, and modification of previous characters: you need to explain why you made this decision. Although I know this is not the case, it could be argued that this is data manipulation with the sole purpose of contradicting the original analysis.

6 >>> We used a new data matrix (a modified version of Ezcurra 2016) because the previous data matrix lacked certain key taxa and characters that we consider important to test the phylogenetic position of a putative early rhynchosaur taxon. Nevertheless, we did run an analysis using the original data matrix, which shows that updating the information for *Colobops* alone is enough to change its phylogenetic position. We include this analysis in the supplementary information and now briefly explain these reasons and actions in the paper. For our first submission, we opted to not include it,

instead only showing and discussing the results using the modified matrix of Ezcurra 2016. We have now added the original analysis and modified the main text (Methods, Results, and Discussion) and supplement accordingly, presenting the data matrix in TNT format and PDF showing the strict reduced consensus tree. Concerning the analysis of the modified matrix of Ezcurra 2016, some character-states of, mostly, lepidosauromorphs and rhynchosauromorphs have been modified based on reinterpretation of the morphology of those taxa, modification of the original formulation of the character or character-states, or new available information. The original character list of Ezcurra (2016) was focused on archosauromorphs and, as a result, its sampling of early lepidosauromorphs and informative characters among these taxa was relatively poor. Thus, we added four species-level lepidosauromorph terminals to sample morphological diversity in the early history of the clade. As a consequence of the taxonomic expansion of the data set, we also added 16 independent, informative characters for our sample of early lepidosauromorphs mostly taken from previous authors (e.g. Pritchard et al. 20018; Simões et al. 2018). We have added information accordingly to the main text.

7 ### Another thing is that there is a recently published dataset for rhynchocephalians, which could be used for a more precise analysis within the group. Maybe the authors would like to take the chance and consider adding a second phylogenetic analysis: Hsiou et al. 2019. A New Clevosaurid from the Triassic (Carnian) of Brazil and the Rise of Sphenodontians in Gondwana (Scientific Reports).

7 >>> A more detailed test of the phylogenetic relationships of *Colobops* within Rhynchocephalia is problematic because the specimen is very likely an early juvenile and we should first explore the ontogenetic variation of phylogenetic characters within the group and their impact on results. Previous authors have demonstrated that early juveniles may have a more basal phylogenetic signal than adults of the same species. Thus, we consider that a detailed, robust phylogenetic analysis of *Colobops* within Rhynchocephalia requires a large amount of research effort that goes beyond the scope of our manuscript (e.g. greater inclusion of characters that document variation among postcranial elements, further inclusion of early lepidosaur taxa, direct examination of specimens, microCT scanning, and careful reassessment of characters that essentially represent the same aspect of morphological variation such as a having a short snout). The specimen also lacks well preserved marginal teeth which are an important source of characters among Rhynchocephalia. Moreover, *Colobops* does not differ substantially from species within *Clevosaurus* with most differences no greater than that found among specimens of the modern *Sphenodon* (exact proportions, precise suture paths, robusticity of particular bones).

Page 9

8 ### Line 14: braincase elements not preserved. From figure 1B, the gray element marked as “?” looks like the postero-lateral portion of the basisphenoid that contributes to the basal tubera, no?

8 >>> it could be, as we state in the descriptive text, but we are not sure and cannot provide more anatomical evidence for identification, therefore we keep it labelled as question mark. Moreover, as preserved the anatomical information that it provides is limited.

Page 12

9 ### Palatal section: a (perhaps supplementary) figure showing a palatal view of the skull without the bones of the skull roof would be helpful.

9 >>> The palatal bones are highly incomplete and damaged (especially the pterygoid) and its identification is only possible in respect to the other cranial elements. We do not see any knowledge gain by adding a figure here.

Page 13

10 ### Line 32: I don't understand what exactly is an artefact. It seems to perfectly match the palatal anatomy of other rhynchocephalians.

10 >>> We agree that the shape in general fits that of other rhynchocephalian taxa. We have reworded the sentence accordingly.

11 ### Lower jaw: a figure of the lower jaw is missing. It could come in the SuppMat, but I would suggest to put it in the main text since some features of the coronoid eminence are discussed elsewhere in the text.

11 >>> The coronoid processes are the only aspects that are preserved of the lower jaw. Both coronoids are adequately visible in figure1 and the reader has access to all the models in the supplement. We thus think an additional figure showing the coronoid processes is not needed.

Page 14

12 ### Line 6: I would avoid the term "ventral emargination". To my knowledge, it is restricted to the ichthyosaur/sauropterygian literature and has a somewhat different anatomical context. I would keep the "ventrally open lower temporal fenestra".

12 >>> The “ventral emargination is not restricted only to Ichthyosauria/Sauropterygia – in reference to an ‘euryapsid’ clade, but is more widely used also for example in descriptions of turtles or parareptile skull anatomy. We use “ventral emargination” as a descriptive term in our paper, without reference to Mesozoic marine reptiles. As such we changed the order of “ventral emargination” and “ventrally open lower temporal fenestra” at first appearance but continue to use ventral emargination as well in the text.

13 ### Skull openings: Since the authors took time to write about them, I would also include the subtemporal fenestra. For instance, it is interesting to notice the ectopterygoid also participates in it... it's not always the case

13 >>> We have added two sentences to address this point.

Page 15

14 #### Line 15: sister taxa instead of closer taxa

14 >>> changed

15 ### Line 27: Why prune *Paliguana* a posteriori to test Bremer supports, instead of a priori?

15 >>> *Paliguana* is a valid taxon and one of the earliest putative lepidosauromorphs. Thus, it is desirable to include its unique combination of character-states in a phylogenetic analysis and not to prune this species a priori. The a posteriori pruning of *Paliguana* still retains the phylogenetic information provided by this taxon in the recovered most parsimonious trees, but it allows testing if its topological instability results in a substantial decrease of decay indices.

Page 19

16 ### Adductor development: first, I would open this section stressing to the reader that the posterior process is incomplete. Second, I would explore more the differences in interpretations for the position of the squamosal, otherwise figure 3 feels mostly unnecessary. So either a good part of figure 3 could be moved to the SuppMat, or the discussion of these differences expanded.

16 >>> we do point out the incompleteness of the posterior process of the postorbital earlier in the manuscript under point “3.7 characterization of adductor development”. Furthermore, as suggested in

another comment by the reviewer, we added to the discussion of the different reconstructions of the jugal in respect to the postorbital (linked to figure 4, not 3). As such, we like to refrain from introducing additional changes to the text.

Page 18

17 ### Line 22: add “in the latter” after “following features” – as it stands, it is confusing if the features discussed refer to Colobops or rhynchosaurs

17 >>> No. The features are found in *Colobops*, not in other non-rhynchosaurid rhynchosaurians. “in the presence of the following features” is the same as “presenting the following features”. We did not change the text here.

18 ### Line 29: Instead of “lateral process”, the authors used “lateral flange” in the description. I would keep consistency.

18 >>> Done

19 ### Lines 31 and 38: is it really possible to state that the features regarding the palatal rami of the pterygoids can be assessed for sure in Colobops? It didn't feel so by the description and images.

19 >>> Yes, based on the symmetry of the remains and the images showing the skull prior to preparation, we are confident in identifying the pterygoid features as described in the text.

Page 19

20 ### Line 43: Why is the left quadrate not segmented or shown in the figures? Or did the authors mean the right quadrate (and right squamosal)?

20 >>> We thank the reviewer for spotting this mistake - in a previous version we had a line on the quadrate here that later should have been completely removed from this list. Left squamosal is correct, but it should have read “compared to the remaining part of the left postorbital”

Page 21

21 ### Lines 25 and 26: the following sentence reads weird in relation to the previous one: “In contrast, the postorbital develops in close contact with the anterior process of the squamosal”.

21 >>> This indeed does not make much sense. We wanted to use “in addition,” – the text has been modified accordingly.

22 ### Line 34 and 35: What is the difference between the two reconstructions? Please make it explicit.

22 >>> We have added some explanation to the text to make this part more clear.

FIGURES

23 ### Fig.1: This figure looks great, but it is a bit confusing. I am not also sure there is a need for 1D-G to be shown in the main text; they could be maybe moved to the SuppMat.

23 >>> We think that all these views reveal aspects of the morphology of the skull and thus would like to refrain from making any changes to the figure.

24 ### Fig.3: Please consider to somehow highlight the name of the taxon in the topology.

24 >>> *Colobops* appears now marked with an asterisk. The caption has been modified accordingly.

25 ### Fig.4: Regardless of being discussed or not in further depth in the “Adductor Development” section, this figure could be reduced and have some views moved into the SuppMat (esp. A). Also, I would invert the left and the right columns to show first the putative position of the squamosal and second the measurements.

25 >>> Figure 4a shows the original condition that was used for measuring by Pritchard et al. 2018 and as such, it is deemed critical as reference for the changes shown in 4b and 4c. We thus like to refrain from moving part of the figure to the supplement. In addition, the images in the left and right are the same models in 4a, 4b, and 4c respectively, just showing different angles to help visualising the changes in position of the squamosal. As such it does not make sense to exchange the left and right images.

Reviewer: 2

Comments to the Author(s)

1 ### My only concern is about the systematic position of Colobops among the Sphenodontia. Could the authors include a more comprehensive phylogenetic analysis with the phylogenetic hypotheses available for Sphenodontia to compare with the actual phylogenetic data of the manuscript? This would help to try to clarify Colobops' closest relationships with other sphenodontian. However, it is not a closed condition, although I believe among the other sphenodontian researchers, this is a primary condition and very important to understanding the Rhynchocephalia evolution.

1 >>> Please see comments of reviewer 1 (7###) and our answers above.